# Isobutanol production freed from biological limits using synthetic biochemistry

Saken Sherkhanov[1], Tyler P. Korman[1,2], Sum Chan [1], Salem Faham[3], Hongjiang Liu[1], Michael R. Sawaya [1], Wan-Ting Hsu[1], Ellee Vikram[1], Tiffany Cheng [1] & James U. Bowie [1✉]

Cost competitive conversion of biomass-derived sugars into biofuel will require high yields, high volumetric productivities and high titers. Suitable production parameters are hard to achieve in cell-based systems because of the need to maintain life processes. As a result, next-generation biofuel production in engineered microbes has yet to match the stringent cost targets set by petroleum fuels. Removing the constraints imposed by having to maintain cell viability might facilitate improved production metrics. Here, we report a cell-free system in a bioreactor with continuous product removal that produces isobutanol from glucose at a maximum productivity of $4\,g\,L^{-1}\,h^{-1}$, a titer of $275\,g\,L^{-1}$ and 95% yield over the course of nearly 5 days. These production metrics exceed even the highly developed ethanol fermentation process. Our results suggest that moving beyond cells has the potential to expand what is possible for bio-based chemical production.

[1] Department of Chemistry and Biochemistry, Molecular Biology Institute, UCLA-DOE Institute, University of California, Los Angeles, CA, USA. [2] Invizyne Technologies, Inc., Monrovia, CA, USA. [3] Vertex Pharmaceuticals, Boston, MA, USA. ✉email: bowie@mbi.ucla.edu

D
ue to increased energy consumption, climate change concerns, and the environmental impact of fossil fuels, there has been an increased demand for sustainable production of biofuels. Living microbes have been engineered to produce a diverse range of potential biofuels, yet it has been difficult to achieve commercially competitive production parameters with live cells[1–7]. Sugar to ethanol conversion represents the current gold standard for biofuels with production metrics of $\sim2\,\mathrm{g\,L^{-1}\,h^{-1}}$, $\sim100\,\mathrm{g\,L^{-1}}$, and $\sim90\%$ yield[8], but most microbial engineering efforts to make other products have fallen far short of these parameters[9,10]. The commodity chemicals 1,3-propanediol and 1,4-butanediol provide perhaps the most successful examples of microbial engineering to date, as they can be generated with similar or better productivity and titer to ethanol, but at yields that are considerably lower than ethanol[11,12]. While the battle continues, the difficulties of fighting life processes in microbial chemical production are well known[4,5,13].

A cell-free approach in which the enzyme pathways are housed in reactors rather than maintained within cells, so-called synthetic biochemistry[14], could free us from biological constraints. Recent advances point to the potential of synthetic biochemistry to achieve remarkable productivities[15,16], near theoretical yields[16–22], and titers that are well above cell toxicity limits[19,23]. It has not yet been demonstrated, however, that the full set of production parameters (titer, productivity, yield) for the cell-free conversion of simple input biomass such as sugar into a next-generation biofuel can approach or surpass ethanol fermentation parameters.

Isobutanol is considered as the next-generation biofuel. It has great advantages over ethanol as a fuel, with lower hygroscopicity, lower volatility, and higher energy density[24]. Moreover, isobutanol has fuel properties comparable to gasoline, current infrastructure is well-suited for its storage and transport, and it can also be converted into jet fuel[24]. There have been numerous efforts to produce isobutanol in cells that are summarized in Supplementary Table 1. To our knowledge, the best-reported metrics for microbial isobutanol production are $22\,\mathrm{g\,L^{-1}}$ titer, $0.7\,\mathrm{g\,L^{-1}\,h^{-1}}$ productivity and a yield of 86% in *E. coli*, although the overall titers can be increased to as high as $50\,\mathrm{g\,L^{-1}}$ by in situ product removal[25,26].

We previously designed a cell-free system for the production of the next-generation biofuel isobutanol[27]. The synthetic biochemistry system we designed is outlined in Fig. 1. As described earlier, the design extends the operational production lifetime by employing an ATP rheostat scheme[27]. The rheostat allows the inevitable losses of ATP energy by hydrolysis to be tolerated (as described previously[27]). The original system achieved $1.3\,\mathrm{g\,L^{-1}\,h^{-1}}$ maximum productivity and reached $24\,\mathrm{g\,L^{-1}}$ in 2 days, with a yield of 91% of theoretical. These isobutanol production metrics already roughly match results reported for cell-based methods so far. While promising, we found that the reaction stopped after several days because two of the enzymes we used in the original effort became inactivated when the titer reached $24\,\mathrm{g\,L^{-1}}$.

Here, we report a major step toward the long-term goal of cell-free chemical production. By improving the solvent tolerance of the enzymes and other process developments, we develop an enzyme system that can further improve isobutanol production from glucose.

## Results

**Enzyme stabilization by genome mining.** Our goal in this effort was to obtain enzyme variants for each of the 16 steps in the pathway, that possess a half-life of a week or more in near saturating isobutanol (8%) at 25 °C. Achieving that goal would greatly mitigate any concerns of isobutanol inactivation limiting titers. We first evaluated the isobutanol tolerance of the 16 enzymes used in the prior work. Only one of the enzymes used

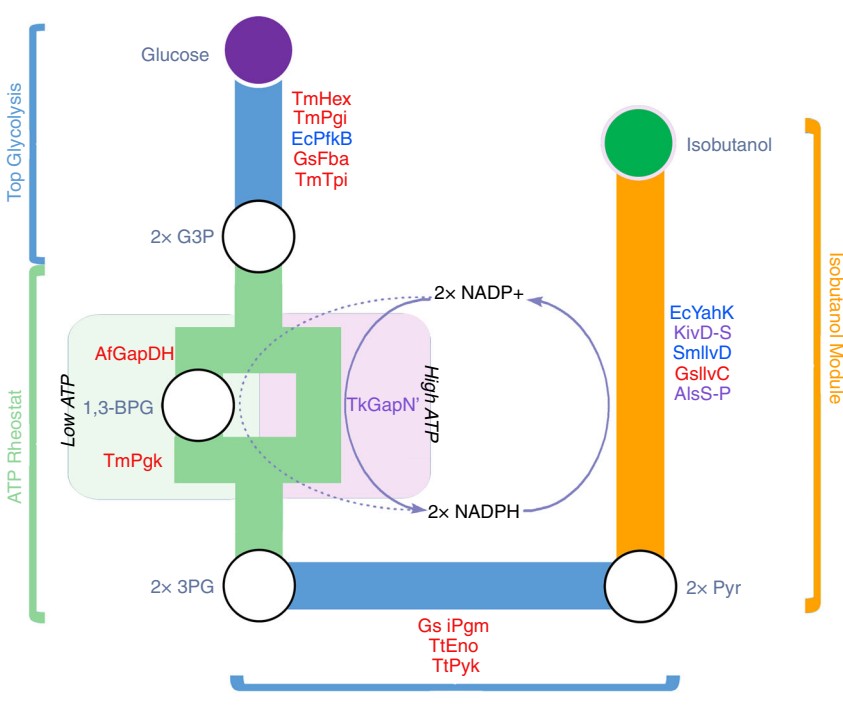

**Fig. 1 Schematic of synthetic biochemistry pathway for conversion of glucose to isobutanol.** Enzymes used in this pathway are described in Table 1. Enzymes in red are from hyperthermophilic organisms, enzymes in blue are mesophilic and enzymes highlighted purple are designed specifically for this study. G3P glyceraldehyde-3-phosphate; 3PG 3-phosphoglycerate; 1,3-BPG 1,3-bisphosphoglycerate; Pyr pyruvate. The logic of the system including the rheostat are described in Opgenorth et al.[27].

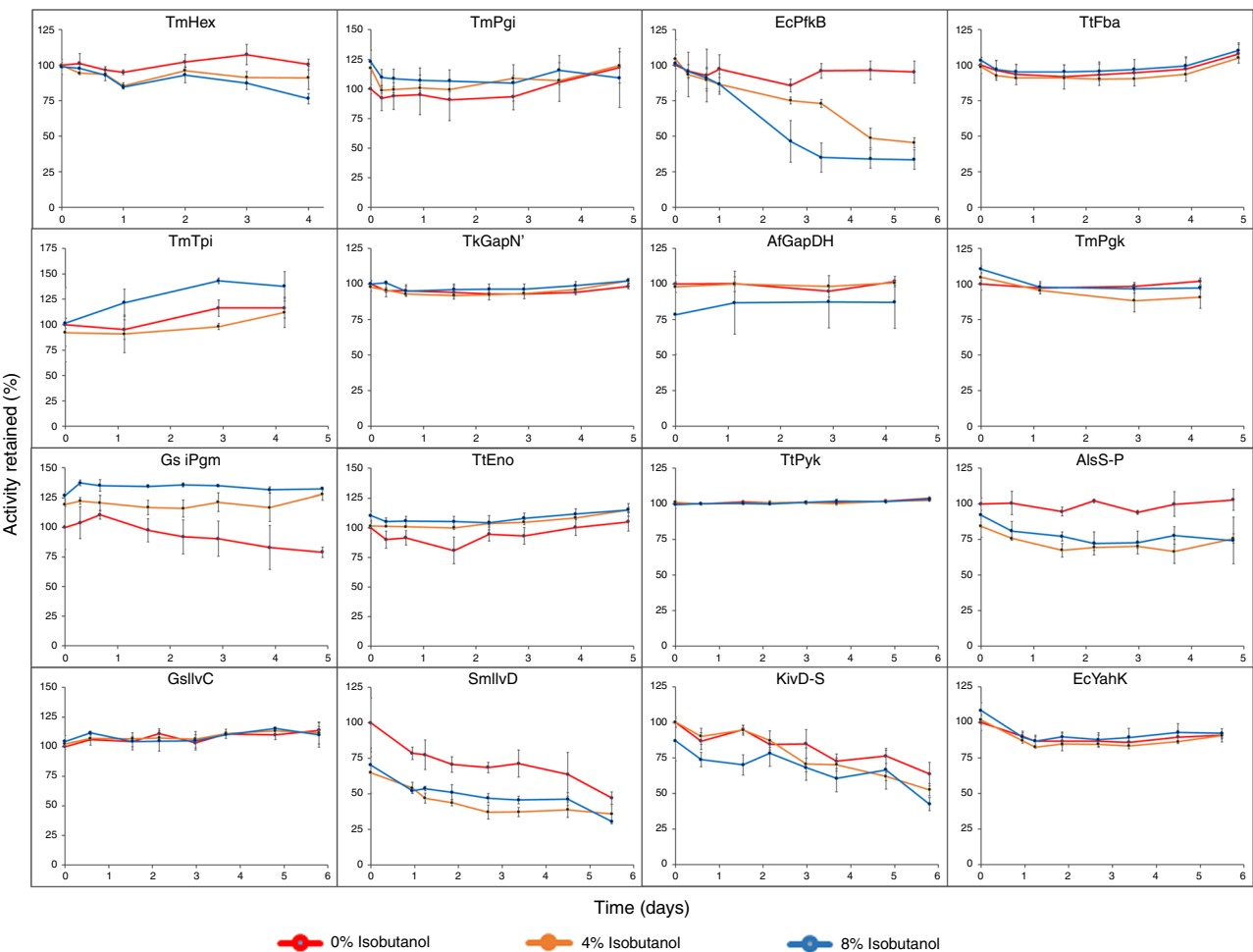

**Fig. 2 Isobutanol tolerance of enzymes.** The indicated enzymes were incubated at 25 °C, in 0% (red), 4% (orange) or 8%(blue) isobutanol. Activity retained is the percentage of the residual activity compared to the initial activity of each enzyme at time zero in 0% isobutanol. The estimated half-life of each enzyme in 8% isobutanol is given in Table 1. Error bars represent the standard deviation of biological triplicates. Source data are provided as a Source data file.

earlier was sufficiently stable in 8% isobutanol: aldehyde reductase from *E. coli* (EcYahK)[27]. As shown in Fig. 2, the activity of YahK is little diminished after 5 days in 8% isobutanol.

For the remaining 15 enzymes, we first employed a genome mining approach, focusing largely on enzymes from thermophilic and hyperthermophilic organisms since thermostability and organic solvent tolerance can be correlated[28]. In addition, thermophilic enzymes are better suited for large scale purification and future industrial implementation, allowing the heat pretreatment of cell lysates to remove impurities[29,30]. In our synthetic biochemistry reactions, however, we employ low temperatures (≤30 °C) to preserve cofactor integrity, so we were concerned that thermophilic enzymes might not be sufficiently active. Thus, our screening criteria for natural isobutanol tolerant enzymes included (1) useable activity at room temperature; (2) high expression in *E. coli* (>10 mg L$^{-1}$); and (3) ~1 week or greater half-life in 8% isobutanol. To find candidate enzymes we used an ad hoc approach. We first searched for homologs, and then to narrow the search where possible, we combed the existing literature to find enzymes known to express in *E. coli*. The enzymes we selected for screening and their properties are listed in Supplementary Table 2. In this manner, we identified 13 additional isobutanol tolerant enzymes (see Table 1 and Fig. 2). While our limited approach was efficient in finding stable and active homologs, it is likely that there are even better natural enzymes that could be discovered in a broader search.

After the genome mining exercise, we were left with two enzymes, acetolactate synthase (AlsS) and α-ketoisovalerate decarboxylase (KivD) requiring stabilization. We also focused on a third enzyme, the non-phosphorylating glyceraldehyde-3-phosphate dehydrogenase from *T. kodakarensis* (TkGapN) which was highly stable, but not NADP$^{+}$ specific at low temperature.

**Stabilizing AlsS.** We chose the AlsS from *B. subtilis* to stabilize because it has a known crystal structure[31] and unlike many other family members[32], it is FAD independent and the oligomer is constructed from a single polypeptide chain. To stabilize AlsS from *B. subtilis*, we employed the PROSS stabilization design server[33] keeping the subunit interface residues fixed (using PDB Code: 4RJK), again speculating that increased thermostability might also increase solvent stability. Indeed, this proved to be true. Based on the PROSS algorithm design suggestions, we prepared variants with 20 mutations plus 3 deletions, 37 mutations plus 3 deletions, and 71 mutations plus 3 deletions. We also prepared the same variants, but without the suggested deletions. Preliminary characterization revealed that the variant with 20 mutations and no deletions (BsAlsS-P) was much more active than the other constructs and was more isobutanol tolerant. As shown in Fig. 2, while the wild-type enzyme is nearly inactive in 8% isobutanol[27], the BsAlsS-P retains ~80% activity over the course of 5 days in 8% isobutanol.

**Table 1 Stable enzyme set.**

| | Enzyme | Organism | Native/Mutant | Activity (units)[b] | Half-life in 8% isobutanol (hours) |
|---|---|---|---|---|---|
| TmHex | Hexokinase[a] | T. maritima | Native | 14.5 ± 0.3 | >>96 |
| TmPgi | Glucose-6-phosphate isomerase[a] | T. maritima | Native | 29.6 ± 1.1 | >>113 |
| EcPfkB | Phosphofructokinase B | E. coli | Native | 8.3 ± 0.1 | 55/123[d] |
| TtFba | Fructose-1,6-bisphosphate aldolase[a] | T. thermophilus | Codon-optimized | 4.7 ± 0.10 | >>117 |
| TmTpi | Triosephosphate isomerase[a] | T. maritima | Native | 210 ± 0.7 | >>100 |
| TkGapN' | Glyceraldehyde-3-phosphate dehydrogenase (non-phosphorylating)[a] | T. kodakarensis | Designed Codon-optimized | 7.6 ± 0.8 | >>117 |
| AfGapDH | Glyceraldehyde-3-phosphate dehydrogenase (phosphorylating)[a] | A. fulgidus | Native | 10.9 ± 0.6 | >>100 |
| TmPgk | Phosphoglycerate kinase[a] | T. maritima | Native | 8.4 ± 0.5 | >>100 |
| Gs iPgm | Phosphoglycerate mutase (2,3-bisphosphoglycerate independent)[a] | G. stearothermophilus | Native | 49.4 ± 2.2 | >>117 |
| TtEno | Phosphoenolpyruvate hydratase[a] | T. thermophilus | Codon-optimized | 117 ± 5 | >>117 |
| TtPyk | Pyruvate kinase[a] | T. thermophilus | Codon-optimized | 23.8 ± 0.8 | >>139 |
| BsAlsS-P | Acetolactate synthase | B. subtilis | PROSS[c] Codon-optimized | 4.9 ± 0.2 | >>115 |
| GsIlvC | Ketol-acid reductoisomerase[a] | G. stearothermophilus | Native | 0.9 ± 0.1 | >>123 |
| SmIlvD | Dihydroxyacid dehydratase | S. mutans | Native | 2.7 ± 0.2 | 125 |
| KivD-S | Alpha-ketoisovalerate decarboxylase | L. lactis | PROSS[c]/DE Codon-optimized | 18.0 ± 1.6 | 123 |
| EcYahK | Aldehyde reductase | E. coli | Native | 4.5 ± 0.4 | 138 |

DE directed evolution.
[a]Enzyme from hyperthermophilic organism.
[b]Unit defined as μmole product min⁻¹ mg⁻¹ enzyme.
[c]PROSS design algorithm[33].
[d]Biphasic inactivation in 8% isobutanol.

**Stabilizing KivD**. In the synthetic isobutanol pathway, KivD is used to decarboxylate ketoisovalerate to form isobutyraldehyde, the penultimate step in the pathway. While there are many homologous decarboxylases, we felt that the KivD enzyme was not a good candidate for the genome mining approach. Many of the homologous decarboxylases that decarboxylate ketoisovalerate will also act on pyruvate, which would derail isobutanol production. Moreover, there are few thermophilic candidates and a previous thermophile prospecting effort[34] failed to identify a suitable replacement for the previously identified, highly specific *Lactococcus lactis* enzyme (LlKivD)[25]. We therefore decided to improve the isobutanol tolerance of LlKivD. Prior directed evolution work had identified a thermostabilized variant of LlKivD called LLM3 which served as a starting point[35], although even the thermally stabilized variant was rapidly inactivated in isobutanol (Supplementary Fig. 1).

While the structure of a fairly close homolog of LLM3 exists (88% sequence identity), we decided to determine the crystal structure of LLM3 so that we could have an accurate model for designed stabilization. LLM3 crystallized in space group $P2_12_12_1$, with four molecules (two dimers) in the asymmetric unit. The structure was refined to 1.8 Å resolution with bound thiamine pyrophosphate ($R_{work} = 18.3$, $R_{free} = 21.0$, PDB ID: 6VGS). Details of the structure determination are described in "Methods" and Supplementary Table 3.

Based on the LLM3 crystal structure (chain A), we employed the PROSS algorithm[33], which suggested three variants, each bearing 21 to 32 designed mutations. We edited the designs to remove mutations we judged as possibly too near the active site. Two of the enzyme designs were inactive, but one bearing 20 mutations after editing was found to be active upon initial purification (KivD-P).

Initial work with KivD-P revealed a complication. KivD-P had the interesting and unusual property of being active upon initial purification, but it slowly became inactive after overnight incubation at 4 °C. The inactive protein could then be reactivated by heating for 15 min at 60 °C, but then became inactivated again after overnight incubation. We don't know the origin of this inactivation/reactivation behavior, but one possibility is that heat drives the protein into an active conformation, which then reverts to an inactive conformation in a slow process with a high activation barrier.

Because KivD-P was clearly more stable than LLM3, we decided to try and eliminate the time-dependent inactivation phenomenon. Starting with KivD-P, we changed each of the 20 mutations back to the wild-type side chain, one at a time, to identify the mutations that were responsible for the slow inactivation property. We found three positions (K64, E95 and S244) that showed slower inactivation when restored to the wild-type side chain so we combined them together to generate a KivD enzyme with 17 mutations (KivD-P2). Kiv-P2 no longer showed the unusual time-dependent inactivation, but retained thermal stability. Although KivD-P2, like LLM3 was rapidly inactivated in 8% isobutanol, we observed greatly improved stability in 4% isobutanol, indicating that Kiv-P2 had better solvent tolerance than the LLM3 starting point (Supplementary Fig. 1). Nevertheless, KivD-P2 was still not sufficiently tolerant.

To improve solvent tolerance further, we turned to directed evolution. In each round of evolution, we screened 1000-3000 mutants for 8% isobutanol tolerance (with each clone bearing on average 2 to 3 randomly introduced nucleotide changes). The first round yielded a mutation T435S. KivD-P2-T435S had ~20% reduced specific activity, but had a slower inactivation rate in 8% isobutanol so it was carried forward into a second round of mutagenesis and screening (Supplementary Fig. 1). Prior to the next round of screening, we added mutation F542L which has been demonstrated to improve KivD activity with substrates larger than pyruvate[36]. KivD-P2-T435S/F542L did not have an obviously improved inactivation rate, but it was more active in 4% and 8% isobutanol at time zero (Supplementary Fig. 1). The second round of screening added two mutations to make KivD-P2-T435S/F542L/A384V/L434M (KivD-S). KivD-S had greatly

improved solvent tolerance, retaining ~50% of its starting activity after 7 days at room temperature in 8% isobutanol (Fig. 2 and Supplementary Fig. 1).

**Altering specificity of TkGapN.** Although a highly specific GapN is not required in the current system since we only employed NADP$^+$, we thought it would provide more flexibility in future pathway designs if we had an NADP$^+$ specific GapN enzyme. For wild-type TKGapN at its optimal temperature of 70 °C, $V_{max}$ is comparable for both NAD$^+$ and NADP$^+$, yet the $K_m$ is much higher for NAD$^{+[37]}$. Thus, at low mM substrate concentrations and optimal temperature of 70 °C, the wild-type TKGapN shows clear preference for NADP$^+$. We found, however, that at low temperature the preference is much smaller. In particular, at 2 mM concentration and 25 °C, the specific activity is only 2.6 ± 0.1-fold higher for NADP$^+$ compared to NAD$^+$. Since we run our system at ≤30 °C, we sought to enhance the NADP$^+$-specificity of TkGapN using structure-guided design. The structure of TkGapN was first modeled using the Phyre2 prediction server[38]. The predicted structure was then superimposed onto the mesophilic streptococcus

mutants GapN (SmGapN, PDB accession: 2QE0)[39], a mesophilic NADP$^+$-specific[40] enzyme with 35% sequence identity to TkGapN. Two residues in contact with the phosphate in the 2′ position of the ribose ring, S199 and S200, were different in SmGapN so we replaced them with the SmGapN residues to make the double mutant S199P/S200T (TkGapN'). TkGapN' did show improved specificity at room temperature with a 11.4 ± 0.1-fold preference for NADP$^+$ at 2 mM substrate concentrations.

**Small scale reactions and possible thermodynamic hurdle.** With an isobutanol tolerant set of enzymes in hand (thermostability information also provided in Supplementary Fig. 2), we next set out to reconstitute the isobutanol production pathway at a small scale. After optimization in 200 μL scale reactions, we were able to approximately reproduce the original system[27] with the new enzymes, reaching a productivity of ~1.3 g L$^{-1}$ h$^{-1}$, a peak titer of 31.5 ± 1.5 g L$^{-1}$ of isobutanol in 25 h with a total enzyme loading of 5 mg mL$^{-1}$ (Fig. 3a). Although the titer with the stabilized enzyme system represents a ~30% increase from the original system (compared to 24 g L$^{-1}$), the improvement was rather

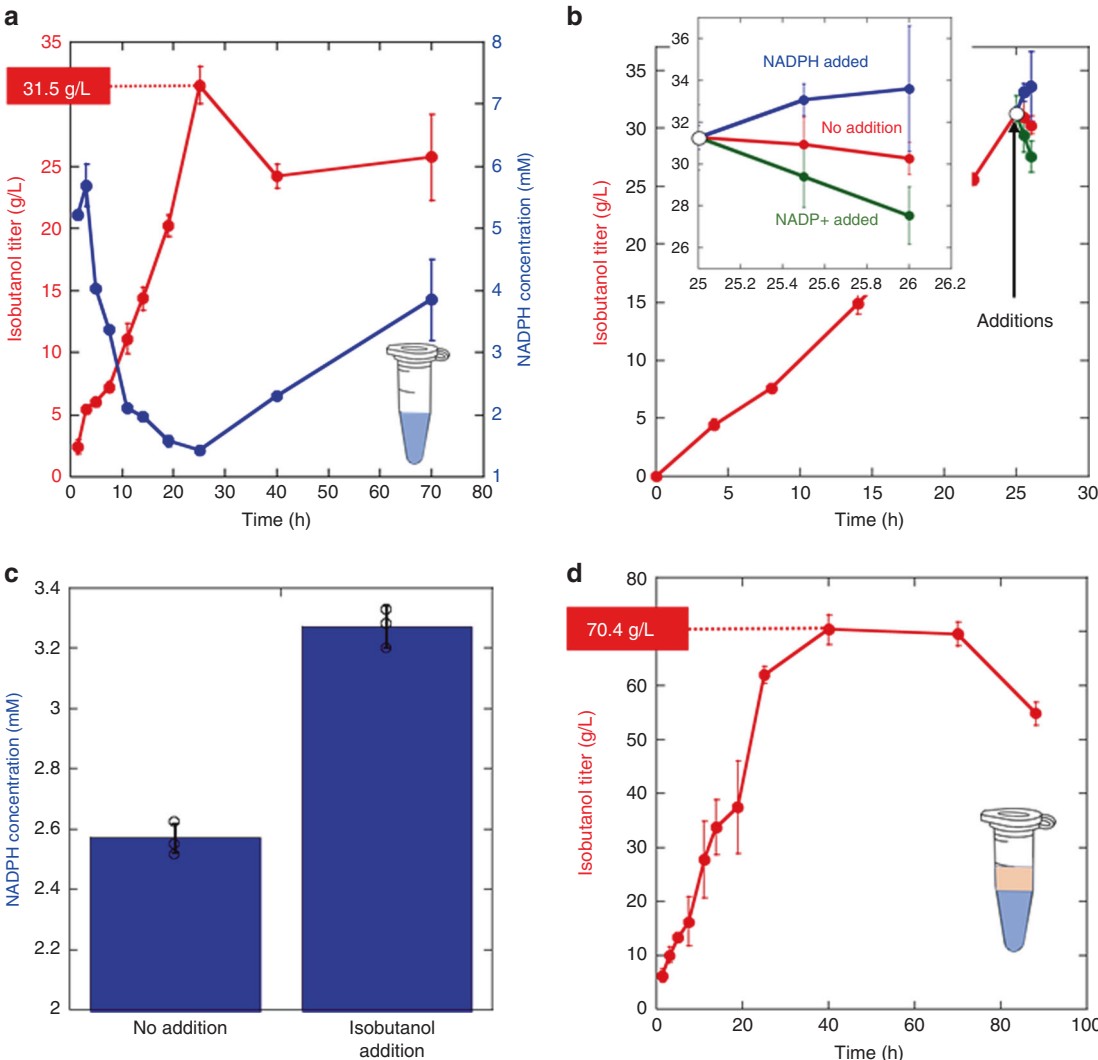

**Fig. 3 Small scale reactions. a** Isobutanol and NADPH titers over time for 200 μL reactions. **b** Effect on isobutanol titers by the addition of 10 mM NADPH or NADP$^+$ at 25 hours, the time at which the reaction achieves approximately maximum titers. The inset is a magnification of the final portion of the graph. NADPH addition raises isobutanol titer, while NADP$^+$ reduces isobutanol titer. **c** The effect of the addition of 100 mM isobutanol at 25 hours on the NADPH levels at 25.5 h. The addition of isobutanol increases the NADPH concentration. **d** Isobutanol titer over time with addition of an organic overlay. Error bars in each panel represent the standard deviation of technical triplicates. Open circles in panel c show the individual data points. Source data are provided as a Source Data file.

disappointing considering the effort we had expended to stabilize the enzymes.

Since all enzymes were isobutanol tolerant, we suspected that there may be another reason for our inability to push to higher isobutanol concentrations. We noticed that after achieving a peak titer, the isobutanol titer unexpectedly retreated to lower levels (Fig. 3a). Moreover, as the isobutanol concentration diminished, we found that the NADPH concentration rose (Fig. 3a). This finding suggested that we may be pushing up against an equilibrium barrier. In particular, the last step in isobutanol biosynthesis is the reduction of isobutyraldehyde:

$$\text{Isobutyraldehyde} + \text{NADPH} = \text{isobutanol} + \text{NADP}^+ \qquad (1)$$

Although this is a highly favorable reaction thermodynamically under standard state conditions ($\Delta G'^o = -24.1\,\text{kJ mol}^{-1}$)[41], as the isobutanol concentration increases, the forward reaction will become increasingly unfavorable (the peak titer of $31.5\,\text{g L}^{-1}$ corresponds to 0.44 M). We tested the possibility that the final equilibrium might be limiting the titer, by first allowing the reaction reach approximately peak titer (25 h), and then adding a bolus of NADPH, NADP$^+$, or additional isobutanol. As shown in Fig. 3b, increasing NADPH concentration boosts isobutanol titer, whereas added NADP$^+$ reduces isobutanol titer. Added isobutanol causes an increase in NADPH levels, suggesting that the increased isobutanol pushes the reaction in reverse (Fig. 3c). The observed responses are expected if the final step is at or near equilibrium.

**Adding an organic overlay**. While it might be possible to drive the reaction forward by further optimization, we decided that it would become increasingly difficult to get to higher titers without a fundamental change in the system itself. We, therefore, chose to modify the process by introducing an organic overlay. We speculated that by using an organic overlay, isobutanol will partition preferentially into the organic layer as it is made, thereby keeping its concentration low enough in the aqueous phase to maintain a sufficient thermodynamic gradient for more isobutanol production. After screening a number of organic solvents, we found that phenetole was particularly effective for isobutanol extaction.

Following the introduction of an organic overlay, we significantly improved the system parameters by doubling the titer to $70.4 \pm 2.7\,\text{g L}^{-1}$ and increased the productivity in the first 24 hours to $2.5\,\text{g L}^{-1}\,\text{h}^{-1}$ (Fig. 3d). The ATP levels after 40 hours of reaction were $2.4 \pm 0.4\,\text{mM}$ and $5.4 \pm 1.1\,\text{mM}$ for systems without and with overlay respectively. The NADPH levels were $2.3 \pm 0.05$ and $1.4 \pm 0.1\,\text{mM}$ for reactions without and with overlay respectively, indicating that the systems were effectively recycling cofactors. The facile introduction of an organic overlay without major concerns about cell toxicity or forming emulsions[42] is a particular advantage of the synthetic biochemistry approach.

**Moving to bioreactor and increasing scale**. In small scale reactions, we noted that the pH drops rapidly due to $CO_2$ release so we sought to improve production parameters further by moving to a bioreactor where we could control pH and temperature. Moving to a bioreactor also required increasing the reaction scale 75-fold to 15 mL. The ability to control pH dramatically improved isobutanol production. Without an organic overlay, the bioreactor system showed increased productivity to $\sim 1.8\,\text{g L}^{-1}\,\text{h}^{-1}$, achieving peak titers of $56 \pm 1\,\text{g L}^{-1}$ in 30 h (Fig. 4). The improved reaction conditions apparently allowed the system to push the apparent equilibrium to higher isobutanol levels. When we then added an organic overlay in the

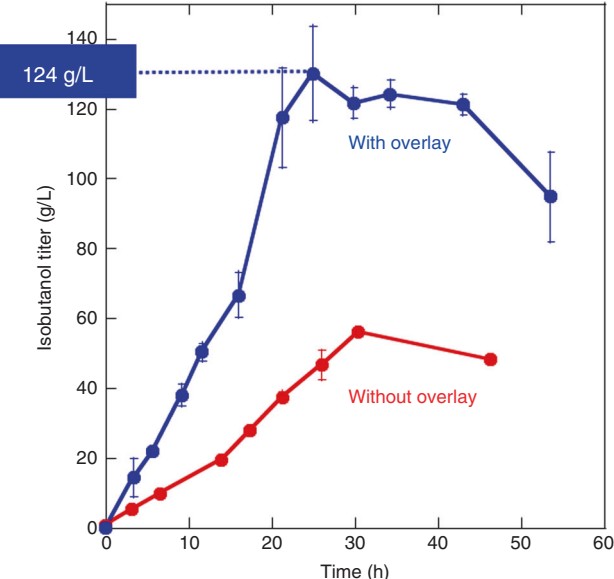

**Fig. 4 Reactions in a bioreactor.** The graph shows isobutanol titers over time for 15 mL reactions in a bioreactor without (red) or with (blue) an organic overlay. Error bars reflect are the standard deviation of technical triplicates. Source data are provided as a Source Data file.

bioreactor, the peak productivity increased to $\sim 5\,\text{g L}^{-1}\,\text{h}^{-1}$ and the titers reached $124 \pm 4\,\text{g L}^{-1}$ in 24 h (Fig. 4). These results further indicate that the cell-free production system scales without difficulty.

**Continuous product removal**. To reduce the buildup of isobutanol that ultimately slows production even with an organic overlay, we next developed a process with continuous removal of the organic layer and collection of the product in a separate container (Fig. 5a). With this set-up, we were able to extend the production period to nearly 60 h with titers reaching $184 \pm 7\,\text{g L}^{-1}$ at a productivity of $\sim 3.2\,\text{g L}^{-1}\,\text{h}^{-1}$ and an overall yield of $91.8 \pm 0.60\%$ (Fig. 5b).

We next asked why the reaction stops at $\sim 60$ h. The residual enzyme activities at 70 hours are shown in Fig. 5c. Most of the enzymes maintain >50% of the starting activity with the exception of KivD and AlsS, which are nearly completely inactivated.

Since the system appeared to remain largely operational with the exception of two enzymes we attempted to extend the life of the system by simply adding a second bolus of KivD and AlsS (easy to do in a cell-free system). We set the system up as before, but this time included a second addition of KivD and AlsS at 33 hours. As shown in Fig. 5d, this approach greatly extended the life of the reaction to $\sim 4.5$ days (108 h). The system achieved a productivity of $4.1\,\text{g L}^{-1}\,\text{h}^{-1}$ for the first 46 hours, a final titer of $275 \pm 3\,\text{g L}^{-1}$ and a yield of $95.4 \pm 0.4\%$ of theoretical.

To learn why the reactions stopped after $\sim 4.5$ days of operation, we again evaluated the enzyme activity and cofactor levels. As shown in Fig. 5d, the ATP levels after 89 h of isobutanol production were $2.6 \pm 0.4\,\text{mM}$ and NAPDH levels when reaction stopped were $1.5 \pm 0.1\,\text{mM}$ indicating that the cofactors were not limiting isobutanol production. There was, however, considerable degradation in the activity of many enzymes, with GapDH, Pgk, Pgm, Eno, Pyk, AlsS, IlvD, and KivD dropping below 20% of their initial activities (Supplementary Fig. 3). We don't know why enzyme half-lives can be notably different in the bioreactor than expected based on isobutanol and thermal tolerance

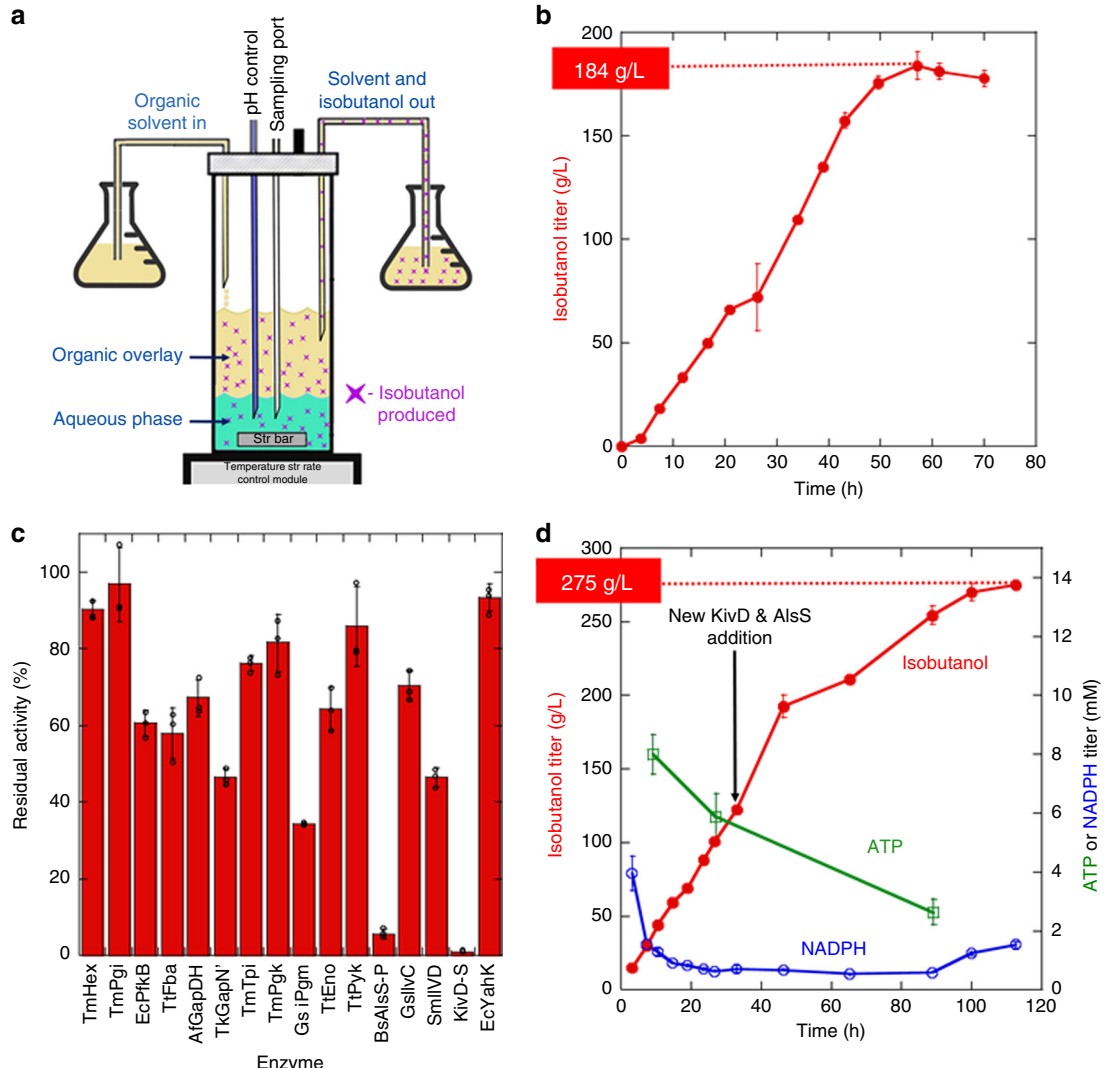

**Fig. 5 Bioreactor system with continuous extraction. a** Bioreactor set up for continuous product extraction. Phenetole in continuously added from a reservoir and simultaneously extracted at the same rate. The extracted layer is collected in a separate container. **b** Isobutanol titer over time using the continuous extraction system. **c** Residual activity of enzymes after 70 hours of incubation. **d** Isobutanol titer over time with the addition of a second bolus of KivD-S and BsAlsS-P at 33 h, the enzymes that become inactivated after 70 h. The ATP (green) and NADPH (blue) concentrations in the reaction are also shown as a function of time. Error bars in all panels are the standard deviation of technical triplicates. Open circles in panel c show the data points. Source data underlying Fig. 5b–d are provided as a Source data file.

measurements for the isolated enzymes. Possibilities include enzyme inactivation at the water-phenetole interface, inactivation by toxic metabolites, or more altered inactivation at high concentrations. Thus, further work remains to learn how to stabilize enzymes under bioreactor conditions, which has the potential to allow the system to reach even higher levels.

## Discussion
Our results show that cell-free systems can generate product continuously at high productivity and yields for many days, achieving production parameters that have so far eluded metabolic engineering despite massive efforts. Moreover, there is much room for improvement. In particular, further investigation of the cause of enzyme inactivation in the reactor could provide pathways to improve enzyme longevity and thereby system sustainability. It also seems possible to achieve even higher productivities by increasing enzyme loadings (currently 5 mg mL$^{-1}$ total). The ease with which we can identify problems (enzyme inactivation,

thermodynamic limitations) in a cell-free system and the ability to flexibly overcome those obstacles (organic layer, enzyme additions) are notable advantages of the synthetic biochemistry approach.

Clearly considerable work still needs to be done before cell-free systems like this would be commercially applicable to such low-value products as biofuels. The major current cost drivers for cell-free biomanufacturing at this time are cofactors and enzymes (see[14,43,44]). It is possible to reduce cofactor costs by employing simpler cofactor analogs such as nicotinamide mononucleotide[44,45], and it may be interesting to explore the use of crude cofactor mixes rather than purified cofactors to lower costs. Methods for cofactor reuse should also be explored. Industrial production of individual enzymes is well developed and, in some cases, can be reduced to a little as $10 per kg[43]. For complex enzyme systems described here, it may be interesting to explore methods for rapidly purifying an entire collection of enzyme activities at once. Highly stable enzyme would allow for extensive enzyme recycling and enzyme immobilization that could greatly

reduce enzyme costs. While only high-value chemicals are currently commercially accessible using enzyme systems as complex as the one described here, as technical developments reduce costs, a wider range of chemicals will become viable. We believe there is considerable potential in the cell-free approach and continued research to explore that potential is warranted.

## Methods

**Materials**. T4 DNA ligase and restriction endonucleases were obtained from New England Biolabs. DNA Polymerase Mastermix was from Denville Scientific. Ni-NTA Superflow, QIAprep Miniprep kits, and QIAquick gel extraction kits were purchased from Qiagen. All reagents and cofactors were from Sigma-Aldrich or Thermo Fisher Scientific. Oligonucleotide primers were synthesized by Valuegene and IDT. Gene sequencing and gene synthesis were performed by Genewiz and Twist Bioscience. Assembly master mix (AMM) used for cloning was prepared as outlined in[19]. All DNA and protein concentrations were measured with a Thermo Fisher Scientific Nanodrop 1000 Spectrophotometer. Vials for small scale reactions, for gas chromatography and 96-well plates were purchased from VWR. Viton tubing was purchased from Cole-Parmer Instrument Company. Colorimetric enzyme-coupled assays were performed in 96-well plates and measured with Molecular Devices SpectraMax M5 microplate reader. When performing large numbers of assays simultaneously, Integra Viafill and Integra Viaflo instruments were used for reagent additions. Colonies for screening of mutant variants were selected automatically using a Molecular Devices Qpix 420 colony picker.

**Cloning and purification of enzymes**. The enzymes used in this study are listed in Table 1. In some cases, we used the natural DNA sequence and in others we employed codon optimization for *E. coli* expression using the IDT server as listed in Table 1. All enzymes were cloned into pET-28a(+) digested with NdeI/XhoI using the Gibson Assembly method and expressed in BL21(DE3) Gold strain. DNA and protein sequences for each enzyme are provided in Supplementary Data 1 and 2.

The enzymes were purified using Ni-NTA affinity chromatography. 1 L of LB media containing 50 μg m$^{-1}$l kanamycin was inoculated with 1 mL of a saturated overnight culture and incubated at 37 °C. When the OD$_{600}$ of the culture reached 0.5–0.7, isopropyl β-D-thiogalactoside (IPTG) was added to a final concentration of 0.5 mM. Strains expressing enzymes from hyperthermophilic organisms were incubated 37 °C with shaking for 18 h (Table 1). The rest of the cultures were incubated at 18 °C with shaking for 18 h. Cultures with cells expressing SmIlvD were supplemented with 0.4 g L$^{-1}$ ferrous sulfate and 0.1 g L$^{-1}$ L-cysteine at the time of induction. The bacterial cells were harvested by centrifugation, resuspended in 50 mL of 50 mM Tris-HCl [pH 7.5], 0.2 M NaCl, and 1 mg mL$^{-1}$ lysozyme, incubated at 4 °C for 30 min with gentle shaking and stored at −80 °C. The frozen bacterial pellet was thawed, lysed by sonication and Avestin Emulsiflex C3 homogenization and cell debris removed by centrifugation at 30,000×g for 40 min. The resulting lysates were treated by either heat (hyperthermophilic enzymes, 65 °C for 3 hours) or isobutanol (mesophilic enzymes, 8% isobutanol for 3 hours at 4 °C). The lysates were reclarified by centrifugation at 30,000 g for 60 min (Table 1). The resulting supernatant was batch bound to 5 mL of Ni-NTA superflow resin at 4 °C for 30 min and loaded into a gravity flow column. The beads were washed 4 times with 10 mL of 50 mM Tris-HCl [pH 7.5], 0.25 M NaCl, 5% glycerol, 5 mM imidazole and the proteins were eluted with 50 mM Tris-HCl [pH 7.5], 0.25 M NaCl, 5% glycerol, 250 mM imidazole. All proteins were dialyzed into 50 mM Tris-HCl [pH 8.0], 20% (v/v) glycerol, 0.2 M NaCl, flash frozen using liquid N$_2$ and stored at −80 °C.

**Enzyme activity and substrate concentration assays**. All measurements of enzymatic activity were performed in 200 μL reactions with 100 mM Tris-HCl pH 7.5, 5 mM MgCl$_2$, 10 mM KCl at 25 °C. Enzymes that consumed or produced NADPH were assayed by monitoring the absorbance at 340 nm. Enzymes that consumed ATP were assayed by monitoring the oxidations of NADH at 340 nm using an ATP regeneration system with PEP and PK/LDH (Sigma-Aldrich). The remaining enzymes were assayed by coupling the reactions to reactions that utilize NADP(H) or ATP. The enzyme assay conditions are listed on Supplementary Tables 4 and 5. Absorbances were monitored using a SpectraMax M5 plate reader. Assays for individual enzymes were initiated by adding enzyme. Enzyme assays in the bioreactor systems were initiated by adding substrate after allowing the assay mixture to equilibrate and consume any residual substrates.

NAPD(H) levels in the isobutanol production reactions were measured by monitoring the absorbance at 340 nm. ATP levels were measured using the ATP Bioluminescent Assay kit from Sigma-Aldrich. Calibration curves for ATP and NADPH were prepared by adding known quantities to the same reaction mixture (metabolites and enzymes) used for isobutanol production, but without any added glucose. Glucose levels were measured by a glucose oxidase assay in 60 mM potassium phosphate pH 5.9, 200 mM MgCl$_2$, 0.15% 4-aminoantipyrine, 0.3% N-Ethyl-N-(2-hydroxy-3-sulfopropyl)-m-toluidine, 150 mM FAD, 15 mM EDTA supplemented with horseradish peroxidase (1.37 mg mL$^{-1}$) and glucose oxidase (0.22 mg mL$^{-1}$). The glucose assay reaction was incubated at 37 °C for 30 min and

the absorbance monitored at 550 nm. Rates were compared to calibration curves developed with known concentrations of glucose to determine concentrations.

**Solvent and thermal inactivation assays**. For the solvent and thermal tolerance assays, the enzymes were incubated with corresponding concentration of iso-butanol or at the indicated temperatures in 50 mM Tris-HCl [pH 8.0], 0.2 M NaCl and 10% (v/v) glycerol and 10 μL aliquots assayed as described above. The enzyme concentrations were such that the 10 μl aliquots contained the enzyme amounts listed in Supplementary Table 5.

**In vitro production of isobutanol from glucose**. Small scale reactions were set up in 2 mL gas-tight glass sealed vials. The reactions, in a final total volume of 300 μl, were composed of 100 mM bis-tris propane pH 8.0, 10 mM KCl, 1 mM glutathione, 12.5 mM MgCl$_2$, 5% glycerol, 0.5 mM MnCl$_2$, 10 mM imidazole, 0.75 mM glucose-1-phosphate, 0.5 mM TPP, 5 mM PO$_4$, 6 mM NADPH, 6 mM NADP, 10 mM ATP and the enzymes at concentrations listed in Supplementary Table 6. The reactions were initiated by adding glucose to a final concentration of 1.5 M and incubated at 30 °C. When using an organic overlay,1 mL phenetole was added.

The bioreactor employed the same conditions as the small scale reactions, but the volume was increased to 15 mL. For the reactions with an overlay, 50 mL phenetole was added on top of aqueous phase. 5.69 g of glucose was added as solid to the enzyme and cofactor master mixture to start the reaction, an additional 5.54 g of glucose was added at 33 hours. A custom-made HEL BioXplorer 5000 bioreactor was employed for 15 mL reactions that allowed for temperature, pH, and agitation control via HEL proprietary software version 2.3.149.1. The pH was automatically maintained above 7.3 using 200 mM NaOH additions. For reactions employing continuous product extraction, the phenetole exchange was performed by inserting Viton tubing touching the phenetole layer and not the aqueous phase (see Fig. 5). Fresh phenetole was pumped into the reactor and the top organic layer was pumped out of the bioreactor at a rate of 2–3 mL h$^{-1}$ using a BioRad Econo-Column Pump.

**Isobutanol quantification**. Samples without the organic overlay were extracted with 3.33 volumes of phenetole. For reactions with the organic overlay, aliquots were removed directly from the phenetole layer. The samples were directly assayed from the phenetole organic layer by injecting 1 μL into a GC-FID (Thermo Scientific Trace 1310) equipped with TG-WaxMS column (0.32 mm × 30 m × 0.25 μm). Helium carrier gas was employed at a flow rate of 30 mL min$^{-1}$. The oven temperature was held at 55 °C for 5 min, raised to 80 °C at 4 °C min$^{-1}$, then to 250 °C at 50 °C min$^{-1}$ and held at 250 °C for 5 min. Both inlet and detector temperatures were kept at 250 °C. For both small scale and bioreactor reactions, a standard curve was prepared with known concentrations of isobutanol sampled in the same manner to quantify the amount of product produced. For reactions with continuous addition and removal of the phenetole layer, we took account of the dilution factor using the following equation:

$$C_i^{expected} = (C_i^{measured}[IN] \times V_B + C_i^{measured}[OUT] \times V_E)/V_B \qquad (2)$$

Where $C_i^{expected}$ is the concentration of isobutanol in the organic layer inside the bioreactor that would have been obtained without any dilution, $C_i^{measured}$ [IN] is the concentration of isobutanol actually measured inside the bioreactor, $V_B$ is the volume of the phenetole layer inside the bioreactor, $C_i^{measured}$ [OUT] is the concentration of isobutanol actually measured in the phenetole removed from the organic layer and $V_E$ is the volume removed.

**KivD LLM3 crystallization and structure determination**. KivD LLM3 was crystallized by hanging drop vapor diffusion. 1.4 μL of 11.8 mg mL$^{-1}$ KivD LLM3 in 20 mM MES, pH 6.8, 2.5 mM MgSO$_4$, 0.1 mM TPP, 1 mM DTT was mixed with 0.7 μL of crystallization reagent (20% w/v PEG 3000, 0.2 M NaCl, 0.1 M HEPES-NaOH pH 7.5) and the drop inverted over the crystallization reagent. Crystals were cryo-stabilized in 25% glycerol/75% crystallization reagent and flash frozen in liquid nitrogen prior to data collection. Data was collected under a stream of liquid nitrogen at APS beamline 24-ID-C. The diffraction data were processed using DENZO and SCALEPACK. Initial phases were obtained by molecular replacement with PHASER, employing chain A of PDB entry 2VBF as a search model, with some high temperature factor moieties deleted. The structure was refined using REFMAC and COOT was used for model building.

**Improving KivD directed evolution**. Mutations were introduced using the Gen-eMorph II Random Mutagenesis Kit (Agilent, catalog number: 200550), along with primers that align upstream and downstream of the KivD insert (DNA primer sequences: 5′-GTGCCGCGCGGCAGCCAT-3′, 5′-GGTGGTGGTGGTGCTCG AGTTA-3′), adjusting conditions so as to obtain 2 to 3 nucleotide changes per gene. Ten clones were sequenced to examine the mutation frequency. The product of the random mutagenic PCR reaction was used as the mega primer for a subsequent PCR reaction with the template plasmid using Phusion high-fidelity DNA polymerase, (NEB catalog number: M0530), to generate plasmids for transformation. The reassembled plasmids were treated with DpnI endonuclease

(NEB catalog number: R0176) for removal of template plasmids before transformation into a BL21-Gold (DE3) *E. coli* expression strain.

From 1000 to 3000 colonies were picked using a QPix 420 colony picker (Molecular Devices) and used to inoculate 160-µL of LB media supplemented with 50 µg mL$^{-1}$ kanamycin in sterile 96-well microtiter plates. Internal controls were incorporated into each 96-well microplate: four wells contained strains expressing wild-type enzyme to establish a baseline for comparison, two cultures were with empty pET28 vector to establish a negative control, and 2 cultures were not inoculated to test for cross contamination. The cultures were shaken overnight at 37 °C at 600 rpm.

To express the enzymes, 5 µL of each overnight culture was transferred to another 160 µL of fresh Terrific Broth media supplemented with 50 µg mL$^{-1}$ Kanamycin in sterile 96-well microplates. The microplates with the remaining inoculum cultures were stored at −80 °C. The microplates with the expression cultures were grown at 37 °C with 600 rpm shaking for 4 to 6 hours until the OD$_{600}$ reached 0.8 to 1.0. The temperature was lowered to 18 °C for 30 minutes before IPTG was added to a final concentration of 0.5 mM and the cultures were incubated overnight at 18 °C.

The cells in microplates were harvested by centrifugation at 3000 × *g* at 4 °C for 20 minutes. The cell pellets were resuspended and lysed enzymatically in 100 µL of a lysis buffer containing 20 mM Tris-HCl [pH7.5], 50 mM sucrose, 1 mM EDTA, 1 mM PMSF, 10 mM β-mercaptoethanol, and 0.05 µL mL$^{-1}$ Ready-Lyse (Lucigen catalog number: R1810M) high-activity lysozyme solution. The cell suspension was shaken at 800 rpm and 25 °C for 20 minutes. 16 µg mL$^{-1}$ *Serratia marcescens* benzonase nuclease in 100 µL of a 40 mM Tris pH 7.5, 200 mM NaCl, 10 mM MgCl$_2$, was added to cell lysate. The lysate was further incubated at 700 rpm at 25 °C for 20 min, and the insoluble fraction pelleted by centrifugation at 3000 × *g* at 4 °C for 30 min.

5 µL of each lysate supernatant was added to 95 µL of buffer containing 100 mM Tris pH 7.5, 8.4% isobutanol, and incubated at 37 °C for 1 to 1.5 h. To assay residual activity, 100 µL of 100 mM Tris pH 7.5, 0.5 mM NADPH, 1 mM TPP, 20 mM ketoisovalerate, 20 mM MgCl$_2$, 0.06 mg mL$^{-1}$ *Clostridium botulinum* YahK-Q100P were added to each reaction. As a control, we assayed 5 µL of the isobutanol-untreated lysate resuspended in 95 µL of buffer containing 100 mM Tris, pH 7.5. Enzymatic activity was measured by monitoring an absorbance at 340 nm. Potential isobutanol tolerant mutants were identified by comparing the treated to untreated activity ratios. The candidates with higher activity were isolated and purified for further analysis.

**Reporting summary**. Further information on research design is available in the Nature Research Reporting Summary linked to this article.

## Data availability

Data supporting the findings of this work are available within the paper and its Supplementary Information files. A reporting summary for this Article is available as a Supplementary Information file. The datasets generated and analyzed during the current study are available from the corresponding author upon request. Coordinates and structure factors have been deposited in the Protein Data Bank under accession code 6VGS. The source data underlying Figs. 2–4 and 5b–d, as well as Supplementary Figs. 1–3 are provided as a Source data file.

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

## Acknowledgements

This work was supported by DOE grants DE-AR0000556 and DE-FC02-02ER63421 to JUB. The work was greatly supported by UCLA-DOE Institute Core Facilities.

## Author contributions

S.S. designed, conducted, and analyzed the bulk of the experiments required to build and evaluate the full isobutanol production system. J.U.B. and T.P.K. developed the overall concept, analyzed data, provided advice, and helped with experimental design throughout. S.F. designed, performed, and analyzed initial systems with thermostable enzymes and pointed out the possibility of a thermodynamic barrier. T.P.K. performed the bulk of the genome mining efforts and S.C. performed all the efforts to prepare KivD-S and BsAlsS-P, with advice from T.P.K. M.R.S. provided key assistance to S.C. in determining the crystal structure of LLM3. L.L. designed and analyzed TkGapN'. W.T., E.V., and T.C. provided expert technical assistance. S.S. and J.U.B. wrote the bulk of the paper with contributions from all the authors.

## Competing interests

J.U.B. and T.P.K. have founded a company Invizyne Technologies to develop cell-free production methods and S.S. is a shareholder. All other authors declare no competing interests.
