## [Peer Review File · Nature Communications]

REVIEWERS' COMMENTS:

Reviewer #1 (Remarks to the Author):

I was reviewer #1 in the previous iteration. The authors have done an excellent job addressing the concerns and comments made. I believe this manuscript is ready for publication and will be widely cited as a pivotal study as the cell-free field continues to develop.

Reviewer #2 (Remarks to the Author):

The manuscript "Isobutanol production freed from biological limits using synthetic biochemistry" described about production of isobutanol by multiple enzymatic reactions without cells. The authors responded to all the issues pointed out in the peer review process [redacted] and it is considered that the current version of the manuscript is suitable for publication in the journal Nature Communications.

Reviewer #3 (Remarks to the Author):

Reviewer:3

Recommendation: Publish after minor revisions noted.

Comments:

As the author commented, so far, the cell-free production of isobutanol is not cost competitive compared to cell-based fermentation and further studies are required for real application (listed in Response to reviewers). However, I agree that this work can be a milestone towards cell-free biomanufacturing and thus I feel that the manuscript should attract much attention to the broad reader of Nature Communications.

Minor point.

In vitro production of isobutanol was performed at 30 °C in both small scale reaction and bioreactor. According to the supplementary Fig.2, the activity of SmilvD gradually decreased at 25 °C and SmilvD showed almost no activity over 37 °C within 12 h. Based on this thermal tolerance results, I expect that SmilvD has low thermal stability and it will lose its activity at 30 °C more rapidly than that at 25 °C. However, according to the Fig.5c, the residual activity of SmilvD is almost 45 %. In case of KivD-S, although it had higher thermal tolerance than SmilvD, it showed almost no activity after 70 hours of incubation (Fig.5c). In addition, AlsS-p also almost lost its activity after 70 hours of incubation (Fig.5c) despite of its high thermal tolerance.

Response to Reviewer Comments

Reviewer #1 (Remarks to the Author):

I was reviewer #1 in the previous iteration. The authors have done an excellent job addressing the concerns and comments made. I believe this manuscript is ready for publication and will be widely cited as a pivotal study as the cell-free field continues to develop.

Reviewer #2 (Remarks to the Author):

The manuscript “Isobutanol production freed from biological limits using synthetic biochemistry” described about production of isobutanol by multiple enzymatic reactions without cells.

The authors responded to all the issues pointed out in the peer review process [redacted] and it is considered that the current version of the manuscript is suitable for publication in the journal Nature Communications.

Reviewer #3 (Remarks to the Author):

As the author commented, so far, the cell-free production of isobutanol is not cost competitive compared to cell-based fermentation and further studies are required for real application (listed in Response to reviewers). However, I agree that this work can be a milestone towards cell-free biomanufacturing and thus I feel that the manuscript should attract much attention to the broad reader of Nature Communications.

Minor point.

In vitro production of isobutanol was performed at 30 °C in both small scale reaction and bioreactor. According to the supplementary Fig.2, the activity of SmilvD gradually decreased at 25 °C and SmilvD showed almost no activity over 37 °C within 12 h. Based on this thermal tolerance results, I expect that SmilvD has low thermal stability and it will lose its activity at 30 °C more rapidly than that at 25 °C. However, according to the Fig.5c, the residual activity of SmilvD is almost 45 %. In case of KivD-S, although it had higher thermal tolerance than SmilvD, it showed almost no activity after 70 hours of incubation (Fig.5c). In addition, AlsS-p also almost lost its activity after 70 hours of incubation (Fig.5c) despite of its high thermal tolerance.

The reviewer raises an important issue as it seems clear that more needs to be done to understand why enzymes inactivate more or less rapidly under the conditions of the bioreactor compared to our test conditions. We discuss this issue at the end of the results section:

“We don’t know why enzyme half-lives can be notably different in the bioreactor than expected based on isobutanol and thermal tolerance measurements for the isolated enzymes. Possibilities include enzyme inactivation at the water-phenetole interface, inactivation by toxic metabolites, or altered inactivation at high concentrations. Thus, further work remains to learn how to stabilize enzymes under bioreactor conditions, which has the potential to allow the system to reach even higher levels.”